# Effects of High Forage/Concentrate Diet on Volatile Fatty Acid Production and the Microorganisms Involved in VFA Production in Cow Rumen

**DOI:** 10.3390/ani10020223

**Published:** 2020-01-30

**Authors:** Lijun Wang, Guangning Zhang, Yang Li, Yonggen Zhang

**Affiliations:** 1College of Animal Science and Technology, Qingdao Agricultural university, No. 700 of Changcheng Road, Qingdao 266000, China; wlj880626@163.com; 2College of Animal Science and Technology, Northeast Agricultural University, No. 600 of Changjiang Road, Harbin 150030, China; zgn1234@126.com (G.Z.); liyang1405053@sina.com (Y.L.)

**Keywords:** metagenomics, fiber decomposition, volatile fatty acid production, fibrinolytic bacteria, gene expression

## Abstract

**Simple Summary:**

The rumen is well known as a natural bioreactor for highly efficient degradation of fibers, and rumen microbes play an important role on fiber degradation. Carbohydrates are fermented by a variety of bacteria in the rumen and transformed into volatile fatty acids (VFAs) by the corresponding enzymes. However, the content of forage in the diet affects the metabolism of cellulose degradation and VFA production. Therefore, we combine metabolism and metagenomics to explore the effects of High forage/concentrate diets and sampling time on enzymes and microorganisms involved in the metabolism of fiber and VFA in cow rumen. This study showed that propionate formation via the succinic pathway, in which succinate CoA synthetase (EC 6.2.1.5) and propionyl CoA carboxylase (EC 2.8.3.1) were key enzymes. Butyrate formation via the succinic pathway, in which phosphate butyryltransferase (EC 2.3.1.19), butyrate kinase (EC 2.7.2.7) and pyruvate ferredoxin oxidoreductase (EC 1.2.7.1) are the important enzymes. The microorganisms are mainly affected by diet and sampling time.

**Abstract:**

The objectives of this study were to investigate the difference in the mechanism of VFAs production combined with macrogenome technology under different forage-to-concentrate ratios and sampling times. Six ruminally cannulated Holstein cows were used in a randomized complete block design. The high forage (HF) and high concentrate (HC) diets contained 70 and 35% dietary forage, respectively. The results showed that pH was affected by sampling time, at 4 h after feeding had lower value. Excepted for acetate, the VFAs was increased with forage decreased. Propionate formation via the succinic pathway, in which succinate CoA synthetase (EC 6.2.1.5) and propionyl CoA carboxylase (EC 2.8.3.1) were key enzymes, and significantly higher in HC treatment than in HF treatment, *Selenomonas*, *Ruminobacter*, *Prevotella*, and *Clostridium* were the main microorganism that encodes these key enzymes. Butyrate formation via the succinic pathway, in which phosphate butyryltransferase (EC 2.3.1.19), butyrate kinase (EC 2.7.2.7) and pyruvate ferredoxin oxidoreductase (EC 1.2.7.1) are the important enzymes, *Prevotella* and *Bacteroides* played important role in encodes these key enzymes. This research gave a further explanation on the metabolic pathways of VFAs, and microorganisms involved in VFAs production under different F:C ration, which could further reveal integrative information of rumen function.

## 1. Introduction

The rumen is well known as a natural bioreactor for highly efficient degradation of fibers, and rumen microbes play an important role on fiber degradation [1], which provides energy and proteins to the host by producing volatile fatty acids and bacterial proteins through anaerobic fermentation [2]. Carbohydrates are fermented by a variety of bacteria and enzymes in the rumen, then transformed into volatile fatty acids (VFAs) [3,4]. These processes are all performed through a series actions of rumen microbial enzymes.

The ratio of acetate, propionate, and butyrate generated in the rumen is affected by the type of forage and the species and quantity of rumen bacteria [5,6]. Meanwhile, sampling time is also an important factor affecting VFAs production [7]. After feeding comparing with before feeding, the concentration of VFAs significantly increased and the pH value significantly dropped in the rumen environment. Previous research has shown that the content of *Fibrobacter* and *Ruminococcus* (both are mainly Fibrinolytic bacteria) increased when dietary fiber increased in the rumen [8]. Zhang et al. (2017) [9] study showed that the forage-to-concentrate ratio had no effect on the content of total volatile fatty acids (TVFAs), but had an effect on the content of acetate and butyrate. Penner et al. (2009) [10] has demonstrated that high concentrate diets decreased ruminal pH and increased VFA concentration, but had no effect on the fractional rate of VFA absorption in vivo. 

Additionally, the majority of previous researches has been conducted on different dietary forage to concentrate ratio, and they only researched the effects of dietary forage to concentrate ratio on VFAs content and microflora [11,12]. In contrast, the effects of different forage-to-concentrate ratios and sampling time on the enzymes involved in VFAs production and the microorganisms that encode these enzymes have received little attention. Nowadays, the more recently developed high-throughput MiSeq sequencing technique has been successfully used to reveal the changes and functions of the ruminal bacteria community [13,14]. An integrated investigation of the composition and abundances of ruminal bacteria could give a better understanding of the microbiota during the transition period of forage to concentrate dietary and before to after feeding. Therefore, this study was designed and carried out to explore the production mechanism of VFAs combined with macrogenome technology under different forage-to-concentrate ratios and sampling times, and identify the key enzymes and the microbes that encode these enzymes.

## 2. Materials and Methods

### 2.1. Ethics Statement

All animal studies were conducted according to the animal care and use guidelines of the Animal Care and Use Committee of Animal Science and Technology College, Qingdao Agricultural University (Qingdao, China).

### 2.2. Experimental Design, Animals Feeding, and Sample Collection

Six ruminally cannulated Holstein cows (BW = 552 ± 38.8 Kg and Age = 3.2 ± 0.70 years-) were blocked into 1 of 2 dietary treatments based on BW and age. Cows were housed in individual tie stalls bedded with wood shavings. The treatments contained 70% (high-forage, HF) and 35% (High-concentrate, HC) dietary forage (dry matter basis), respectively. For 3 weeks before sampling, animals were fed once daily at 8:00 a.m. and allowed ad libitum consumption of 110% of their expected intake. The ingredient and nutritional composition of the two diets are presented in Appendix A. Rumen content samples were collected before feeding (i.e., at 0 h, BF0 h) and 4 h after feeding (AF4 h) via a ruminal fistula. Samples for DNA and RNA extraction were snap frozen in liquid nitrogen and stored at −80 °C until analysis. Collective representative samples (the sample of each cow taken from all around of the rumen, and then mixed) for VFAs assays were extruded through four layers of cheesecloth. Freshly prepared metaphosphoric acid (25% w/v; 1 mL) was added to 5 mL of filtered rumen fluid and stored at −20 °C for the measurement of VFAs.

### 2.3. Sample Measurements

For the determination of VFAs, samples with metaphosphoric acid were thawed at room temperature and then centrifuged (12,000× *g* for 15 min at 4 °C). The supernatant was used to measure the VFAs. The VFA concentrations were determined by a capillary column gas chromatograph [15]. 

The lactate content in the samples was determined with high-performance liquid chromatography (HPLC, Waters 600, Milford, Massachusetts, USA) using the Agilent HC-C18 column (Bio-Rad, Palo Alto, Calif, USA) and a refractive index detector (Waters 2414) with 9:1 NaH_2_PO_4_ and H_3_PO_4_ as the mobile phase, a column temperature of 60 °C, and a velocity of 1.0 mL∙min^−1^, as assessed by a refractive index detector.

### 2.4. RNA Extraction, RNA Reverses Transcribed, qPCR Primer Design and Analysis

RNA extraction was performed using the liquid nitrogen grinding method and TRIzol reagent (Ambion, Carlsbad, CA, USA) following the protocols described by Kang et al. (2009) [16] with some modifications. The RNA was reverse-transcribed into cDNA using a PrimeScript™ 1st strand cDNA Synthesis Kit (Code No. 6110A, Takara, Dalian, China), following the kit instructions. The reverse-transcribed PCRs were conducted as follows: 37 °C for 15 min, 85 °C for 5 s, and 4 °C for 10 min. The cDNA was stored at −80 °C. The PCR primers used are listed in Appendix A and were assembled based on previous literature [17,18]. Primers were provided by Sangon Biotech Co. Ltd. (Shanghai, China). The Real-Time qPCR performed using Takara SYBR^®^ Premix Ex Taq™ Synthesis Kit (Code No. RR420A, Takara, Dalian, China), following the kit instructions. The abundance of these microbes was expressed as a proportion of total estimated rumen bacterial 16S rDNA according to the equation: relative quantification = 2^−(Ct target-Ct total bacteria)^, where Ct represents threshold cycle [19].

### 2.5. DNA Extraction, Metagenome Library Preparation and Sequencing

Genomic DNA was extracted as described by Minas et al. (2011) with some modifications. DNA was extracted using the Cetyltrimethylammonium Ammonium Bromide (CTAB) based DNA extraction method. The CTAB lysis buffer contained 2% w/v CTAB (Sigma-Aldrich, Poole, UK), 100 mM Tris-HCl (pH 8.0), 20 mM EDTA (pH 8.0) and 1.4 M NaCl (Aladdin, Shanghai, China). The pH of the lysis buffer was adjusted to 5.0 prior to sterilization by autoclaving [20]. The final DNA was resuspended in 50 μL of TE buffer (10 mM Tris-HCl, pH 8.0, and 1 mM EDTA, pH 8.0) and stored at −80 °C.

Qualified DNA samples were first cut into smaller fractions by nebulization. Then, using T4 DNA polymerase, the Klenow fragment and T4 polynucleotide kinase convert the fragmentation-produced overhang into blunt ends. After the adenine (A) base was added to the 3′ end of the blunt-ended phosphorylated DNA fragments, the adaptor was ligated to the end of the DNA fragment. Ampure beads were used for purification and elimination the short fragments. PCR amplification were performed to enrich the adapter-ligated DNA fragments. Then, the PCR products were purified with an AxyPrep Mag PCR clean up kit (Axygen, Corning, NY, USA) following the manufacturer’s recommendations. Sample libraries were quantified and analyzed using the Agilent 2100 Bioanalyzer and the ABI StepOnePlus Real-Time PCR system. The qualified libraries were then sequenced on the Illumina HiSeq™ platform.

### 2.6. Metagenome Assembly, Taxonomic and KEGG Analysis

SOAPdenovo2 was used to reassemble high-quality data [21]. SOAPdenovo results were further assembled with Rabbit to obtain longer contigs [22]. For each sample, the reads were assembled in parallel with a series of different k-mer sizes. SOAP2 was used to map the reads back to each assembly result, and selected the optimal k-mer size and assembly results based on contig N50 and mapping rate [23]. Based on the assembly results, MetaGeneMark v2.10 [24] using default parameters (http://exon.gatech.edu/GeneMark/metagenome/Prediction/) predicted the presence of open reading frames. Genes from different samples were combined by CD-Hit clustering [25] (sequence identity threshold, 95%; alignment coverage threshold, 90%).

The enzyme results of the gene were searched against the sequences in the NR database using the blastp algorithm with an E-value cutoff of 1 × 10^−5^, the best hits were subjected to analysis with Metagenome Analyzer (MEGAN) [26], a program for taxonomic analysis, which could accurately classify DNA sequences as short as 100 bp.

For functional analysis, classification functions were classified using the KEGG orthology database (version 67.1) to identify relationships between various pathways and obtain KEGG numbers and EC numbers. First, we matched the reads directly to KEGG genes, and the mismatch is allowed to be within 10%. All KEGG Orthologue groups (KO) with a hit equal to the best hit were examined. If we were unable to resolve the read to a single KO, the read is ignored; otherwise, the read was assigned to a unique KO. Statistical analysis was performed on KO reads using the PROC MIXED of SAS 9.4 (SAS Institute Inc., Cary, North Carolina, USA).

### 2.7. Statistical Analysis

Data of VFAs, pH, lactate, Pyruvate, microbial diversity and the result of Real-time PCR quantification were analyzed as a randomized complete block design using the PROC MIXED procedure of SAS 9.4 (SAS Institute Inc., 2001), which included feed, time, and feed × time as the fixed effects, and individual group as the experimental unit. Statistical significance was declared at 0.01 < *p* < 0.05 (*) and *p* < 0.01 (**).

## 3. Results 

### 3.1. Rumen Fermentation Parameters of Cows Feed Two Different Diets

An overview of the analyses of in vivo fermentation (Table 1), the content of rumen acetate, propionate, butyrate and total volatile fatty acids (TVFAs) were significantly affected by forage-to-concentrate ratio. The content of acetate and the ratio of acetate:propionate (A:P) showed a significant increase with the dietary forage level increase (*p* < 0.01), while the content of TVFA, propionate, butyrate significantly decreased (*p* < 0.01). The content of propionate was significantly higher in the HC diet and after feeding, and the propionate, pH, and A/P were also affected by the interaction of feed and time. Time (before and after feeding) is also an important factor affect the pH, acetate, propionate, butyrate, and TVFA. The concentration of propionate and butyrate significantly increased with increase in the proportion of dietary concentrate, which led to an increase in total VFA concentration, and a significant decrease in the ratio of A/P, which changed the pattern of rumen fermentation.

### 3.2. Genes Directly Involved in Propionate

Many genes involved in the metabolism of short-chain fatty acids are diverse and abundant in the metagenomic dataset. Among them, genes encoding pyruvate fermentation to propionic acid include the acrylate pathway and the succinate pathway. Plant polysaccharides are fermented by the rumen microbial, and finally, which is mainly production propionate.

Genes encoding enzymes that are directly involved in propionate were analyzed for their abundance in HC and HF treatments. Genes encoding the lactate-acrylate pathway produce propionate (Appendix A and Appendix A). Diet had a significant effect on the genes with only the exception of gene K00016 for acyl-CoA dehydratase (EC:1.3.8.7). The K00249 for lactate dehydrogenase (EC:1.1.1.27) was higher in the HF treatment than that in the HC treatment, whereas, K01026 for lactyl transferase (EC:2.8.3.1) was just the reverse. However, in the lactate-acrylate pathway, the representative genes of lactoyl-CoA dehydratase (EC: 4.2.1.54) are not included in the KEGG gene dataset; therefore, this metabolic pathway cannot produce propionate. 

Conversely, fermenting to propionate seems to be more likely given the high readings of the genes involved via the succinate pathway. Such a pathway has been demonstrated in Figure 1. Genes involved in the succinate pathway to propionate, genes K01958, K01959 and K01960 for pyruvate carboxylase (EC 6.4.1.1) and genes K01902 and K01903 for succinate-CoA synthetase (EC:6.2.1.5) were affected by dietary treatment, and a significant interaction between diet and time was observed for Gene K01026 for propionate CoA transferase (EC:2.8.3.1) (Table 2). Gene K00024 for malate dehydrogenase (EC:1.1.1.37) and genes K01847, K01848, and K01849 for methylmalonyl CoA mutase (EC:5.4.99.2) were both affected by time, and after feeding were higher compared to before feeding. Succinate-CoA synthetase (EC:6.2.1.5) was affected by both feed and time, and HC4h was highest among the four groups (Table 2). The other intermediate enzymes were all found, and no differences were observed among them.

Succinate-CoA synthetase (EC 6.2.1.5) and propionate CoA transferase (EC 2.8.3.1) are the key enzymes in the succinate pathway of propionate formation. Moreover, it was found that the content of the two enzymes was low, so they were still the rate-limiting enzymes in the succinate pathway of propionate formation. 

*Bacillus*, *Prevotella*, *Ruminobacter*, *Selenomonas*, and *Vibrio* were the main microorganisms encoding Succinate-CoA synthetase (Figure 2A), indicating that such microorganisms played a major role in the propionate formation, and the relative content in HC group was significantly higher than that in HF group. Propionyl-CoA transferase encoding by Clostridium and unclassified bacteria in HF group, but only unclassified bacteria in HC group (Figure 2B), indicating that Clostridium plays a major role in the propionate formation.

### 3.3. Genes Directly Involved in Butyrate 

Many genes involved in butyrate metabolism were differentially abundant in the metagenome datasets among the four groups (Figure 3 and Table 3). With the exception of gene K00074 for 3-hydroxybutyryl-CoA dehydrogenase (EC:1.1.1.157) and genes K01715 for enoyl-CoA hydratase (EC:4.2.1.17), all genes encoding the translation of pyruvate into butyrate showed different in the two treatments or before and after feeding. The genes K00169 and K00190 for pyruvate ferredoxin oxidoreductase (EC: 1.2.7.1) was affected by an interaction between feed and time. The effect of time was observed in acetyl-CoA acetyltransferase (EC:2.3.1.9) between the before and after feeding, and acetyl-CoA acetyltransferase was higher after feeding than before feeding. Gene K00248 for butyryl-CoA dehydrogenase (EC:1.3.8.1) and genes K01034 and K01035 for acetate-CoA transferase (EC:2.8.3.8) were affected by feed and showed higher in HF treatment compared to those in HC treatment. The enzymes of butyryl-CoA dehydrogenase (EC:1.3.8.1) and acetate-CoA transferase (EC:2.8.3.8) were needed for the last two steps in the formation of butyrate, Therefore, the butyrate in the HF treatment should be higher than that in the HC treatment, which was consistent with the content of butyrate. Gene K00929 for butyrate kinase (EC:2.7.2.7) and gene K00634 for phosphate butyryltransferase (EC:2.3.1.19) were affected by both feed and time. From Figure 3, we can see that butyrate kinase (EC:2.7.2.7) is the final enzyme to produce butyrate from another branch pathway, and HC4h was highest, which was consistent with the content of butyrate, indicating butyrate via butyrate kinase pathway formation. However, the genes for acetoacetyl-CoA reductase (EC:1.1.1.36) and correlations to butyrate yield were not available in the KEGG database. Therefore, butyryl-CoA directly produces butyrate via acetate-CoA transferase (EC: 2.8.3.8), rather than another branch pathway. This result may be due to the high content of rumen microbes that produce butyrate or use of acetate as a precursor for butyrate synthesis under conditions of lactate fermentation (where lactate fermentation could occur).

Phosphate butyryltransferase (EC 2.3.1.19), butyrate kinase (EC 2.7.2.7) and Acetate-CoA transferase (EC: 2.8.3.8) are key enzymes in the butyrate formation. At genus level, the microorganisms encoding phosphobutyryltransferase and butyrate kinase were mainly *Prevotella* and *Bacteroides* (Figure 4A,B). *Prevotella* was significantly higher in HC group than in HF group, which indicated that *Prevotella* promoted the production of butyric acid in a high concentrate diet. Acetate-CoA transferase is mainly encoded by *Oscillibacter* and *Porphyromonas* (Figure 4C).

### 3.4. The Major Microbial Material Involved in Cellulose and Volatile Fatty Acid Metabolism Gene Characterization and Quantification

An abundance of the predominant microorganisms involved in cellulose and VFA metabolism were measured by quantifying the bacteria gene copy number as well as the expression level in two treatment and before or after feeding. In our study, we selected eight representative bacteria (Table 4). A significant (*p* < 0.05) interaction between feed and time was observed for all bacteria, with the exception of *Fibrobacter succinogenes* and *Megasphaera elsdenii*. *Ruminococcus albus* and *R. flavefaciens* are considered to be the major cellulolytic bacteria in the rumen [27], and the abundances of *Ruminococcus albus* and *R. flavefaciens* 4 h after feeding was higher than that at 0 h before feeding. The relative content of main cellulolytic bacteria *Fibrobacter succinogenes* in HF treatment was significantly (*p* < 0.01) higher compared to that of the HC treatment. The relative content of the species *Prevotella ruminicola*, *Selenomonas ruminantium*, and *Megasphaera elsdenii* were higher in the HC group than in the HF group, of which *Selenomonas ruminantium* was also significantly (*p* < 0.01) higher after feeding than before feeding. These results confirmed the previous research in this study on cellulose and enzyme activity.

## 4. Discussion

In the present study, we investigated the fermentation parameters, the amount of TVFA produced is relatively low of HF treatment, therefore, the fermentation efficiency of forage in the rumen is relatively low. As the proportion of forage diets increases, the proportion of acetate rises and propionate decreases [28]. VFA is the main factor that directly affects rumen pH. VFA absorption in the rumen is a passive process [29], the VFA transfer from the luminal to the surface of epithelial cells through the rumen movement, while increase in the proportion of forage diets increases the rumen movement, and decreases the lumen VFA, thereby significantly increasing the rumen pH. This argument supports the results of this experiment. In regard to the inverse relationship between A:P and concentrate dietary, early studies suggested that fiber-degrading bacteria produced acetate, and starch-decomposing bacteria produced propionate [30]. However, subsequent pure culture studies did not support this view. Fiber-degrading bacteria *R.albus* produced a large amount of acetate, while *R.flavefaciens* and *F.succinogenes* produced a large amount of succinate, which eventually converted to propionate [31,32]. Some starch-decomposing bacteria produce succinate or propionate, but more also produce acetate [33].

VFAs act as energy sources that could meet the production performance of ruminants, therefore, it was important to study the metabolism of VFAs. The production of acetate was contained in the metabolic pathway of butyrate; therefore, the metabolism of propionate and butyrate was the main research focus.

In the rumen, plant polysaccharides are fermented by the rumen microbes, propionate is one of the final productions. Propionate can be produced by acrylate pathway and succinate pathway. The acrylate pathway is the primary pathway in the case of animal diets with high starch content [34], and *Megasphaera elsdenii* is the major propionate producer via the acrylate pathway [35]. It is worth noting that the content of *Megasphaera elsdenii* was very low at the expression level (Table 4) and genes for lactyl-CoA dehydratase were absent. Therefore, fermenting to propionate seems to be more likely given the high readings of the genes involved via the succinate pathway. In the succinate pathway, except for the last enzyme propionate-CoA transferase (EC:2.8.3.1), the succinate-CoA synthetase (EC: 6.2.1.5) was lower compared to the other enzymes in succinate pathway. Propionate-CoA transferase (EC:2.8.3.1) was the last step enzyme, it was the limit-enzyme in the succinate pathway, and was highest in HC-AF4h group than the other three groups. Although this result was consistent with propionate content (Table 1), but low propionate-CoA transferase (EC:2.8.3.1) was not sufficient to affected the propionate content. Subsequently, propionate is also produced by decarboxylation of succinate by *Selenomonas ruminantium* [36]. *Selenomonas ruminantium*, another function, can also produce propionate by fermenting carbohydrates or lactate [36]. In this study, the expression of *Selenomonas ruminantium* in HC treatment was higher compared to in HF treatment, and it was also affected by sampling time, 4 h after feeding was higher than 0 h before feeding. The results of expression of *Selenomonas ruminantium* (Table 4) was consistent with the content of propionate (Table 1), this result further confirmed that the propionate produced mainly by decarboxylation of succinate. 

The content of butyrate in the HC treatment was higher than that of the HF treatment, and the 4 h after feeding group was higher than that at 0 h before feeding, this result was consistent with that of Ribeiro et al. (2015). Acetate and butyrate can be converted into each other in the rumen. Previous study has shown that 28% of acetate in the rumen is not absorbed by the rumen in the form of acetate [37]. Meanwhile, microorganisms can use acetate to produce butyrate by acetyl-CoA transferase and/or butyryl-CoA transferase [38]. Under a high-concentrate diet, microbes in the rumen can promote the metabolism of microorganisms through this energy dissipation process, and continuously convert acetate into butyrate [38]. Lactate-producing bacteria such as *Butyrivibrio fibrisolvens* also can directly produce butyrate using acetate via butyryl-CoA/acetic acid-CoA transferase (EC:2.8.3.8) rather than converting two acetyl-CoA to acetoacetyl-CoA [39]. At the transcriptional level, the relative content of *Butyrivibrio fibrisolvens* in the HC treatment was higher than that in HF treatment, which was similar to the findings of Diezgonzalez et al., (1999). On the other hand, *M. elsdenii* produces butyrate via the malonyl-CoA pathway from various reactions involving acetyl-CoA, which is activated by acetate and is combined with CO_2_ to form malonyl-CoA [35]. Our recent work indicated that the relative content of *M. elsdenii* was very low in both HF and HC groups; however, under this condition, fermentation to butyrate seems more likely via the butyryl-CoA/acetate-CoA transferase pathway using acetate as an acceptor.

## 5. Conclusions

In conclusion, this study combined metagenomics and metabolism to explore the variations in the enzyme activity and the rapidly changing bacterial microbiota in response to the change in diet and time during cellulose to end product VFAs in rumen. Feeding the HF diet increased ruminal pH and decreased TVFA concentration compared to HC treatment. Propionate formation via the succinic pathway, in which succinate CoA synthetase (EC 6.2.1.5) and propionyl CoA carboxylase (EC 2.8.3.1) were key enzymes. Butyrate formation via the succinic pathway, in which phosphate butyryltransferase (EC 2.3.1.19), butyrate kinase (EC 2.7.2.7) and pyruvate ferredoxin oxidoreductase (EC 1.2.7.1) are the important enzymes, and *Prevotella* promotes butyrate production in high concentrate diets and *Bacteroides* promotes butyrate production in high fiber diets. A greater understanding of mechanisms involved in VFAs production and metabolism will increase our knowledge of fiber degradation in cow rumen and may lead to the development of strategies to increase the efficiency of fiber degradation.

## Figures and Tables

**Figure 1 animals-10-00223-f001:**
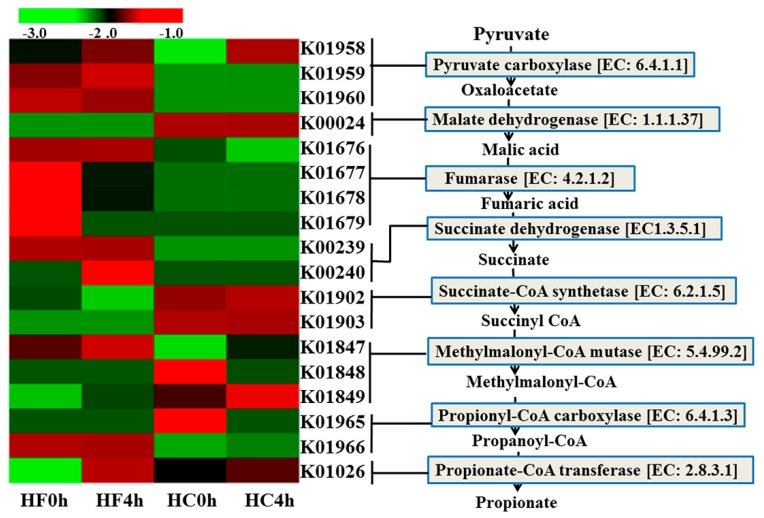
The metagenomic abundance of key elements of the propionate production pathway. Left pane: heat map of KEGG orthologues for the EC numbers involved in propionate production (lines connect the heat map to the propionate production pathway indicating which K0 numbers represent the given enzymes). Right pane: the propionate production pathway showing enzyme classification (EC) numbers. Green and red color indicates low and high abundance, respectively. Computation based on Log10.

**Figure 2 animals-10-00223-f002:**
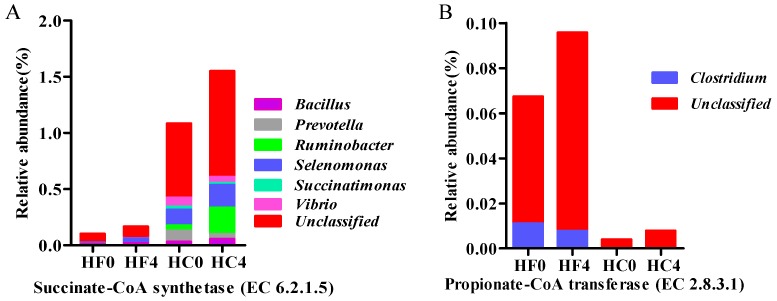
Microbial composition of key enzymes encoding the propionate production pathway (Genus); (**A**): Sucinate-CoA synthetase (EC6.2.1.5); (**B**): Propionate-CoA transferase (EC2.8.3.1).

**Figure 3 animals-10-00223-f003:**
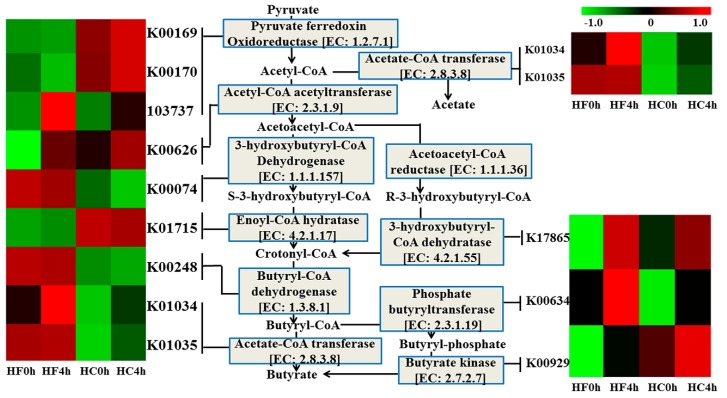
The metagenomic abundance of key elements of the butyrate production pathway. Centre pane: the butyrate production pathway, plus ancillary reactions, showing enzyme classification (EC) numbers. Left and right pane: heat map of KEGG orthologues for the EC numbers involved in butyrate production (lines connect the heat map to the butyrate production pathway, indicating which K0 numbers represent the given enzymes). Green and red color indicates low and high abundance, respectively. Computation based on Log10.

**Figure 4 animals-10-00223-f004:**
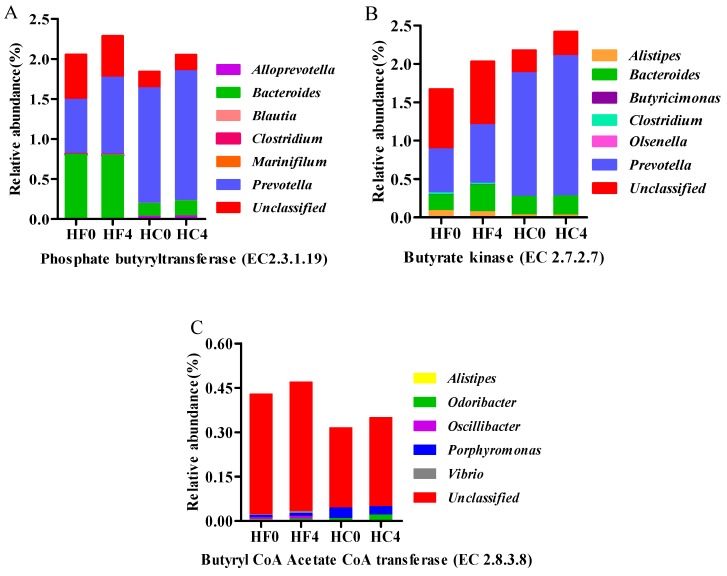
Microbial composition of key enzymes encoding the butyrate production pathway (Genus). (**A**): Phosphate butyryltransferase (EC2.3.1.19); (**B**): Butyrate kinase(EC2.7.2.7); (**C**): Butyryl CoA Acetate CoA transferase (EC2.8.3.8).

**Table 1 animals-10-00223-t001:** Ruminal fermentation characteristics of cows fed two different diets.

Item	HF		HC		SEM ^5^	*p*-Value ^6^		
	BF0h ^3^	AF4h ^4^	BF0 h	AF4 h		Feed	Time	Feed × Time
pH	6.90	6.06	6.84	5.76	0.01	NS	**	*
Acetate, mM	79.84	111.69	69.24	92.94	24.70	**	**	NS
Propionate,mM	19.32	23.87	27.12	49.20	5.76	**	**	**
Butyrate, mM	9.73	13.56	10.51	18.76	3.16	**	**	*
TVFA ^1^, mM	108.89	149.12	106.87	160.90	52.97	**	**	NS
A/P ^2^	4.21	4.67	2.56	1.91	0.04	**	NS	**
Lactate (mmol/L)	7.27	5.40	8.48	6.61	0.86	NS	**	NS
Pyruvate (μg/L)	263.46	290.68	313.60	282.49	131.15	*	NS	*

^1^ TVFA: Total volatile fatty acid; ^2^ A/P: Acetate/ Propionate; ^3^ BF(0), before feeding (0 h); ^4^ AF(4), after feeding (4 h); ^5^ SEM for feed × time; ^6^ NS, not significant (*p* > 0.05); *, (*p* < 0.05); **, (*p* < 0.01).

**Table 2 animals-10-00223-t002:** The percentage of reads mapped to each enzyme involved in propionate production.

Enzyme	HF		HC		SEM ^3^	*p*-Value ^4^		
	BF0h ^1^	AF4h ^2^	BF0h	AF4h		Feed	Time	Feed × Time
EC:6.4.1.1	3.1801	3.6213	2.7825	2.6925	0.0456	**	NS	NS
EC:1.1.1.37	2.6938	3.3037	2.8694	3.3698	0.0712	NS	*	NS
EC:4.2.1.2	3.5578	4.1535	3.6002	3.7724	0.1473	NS	NS	NS
EC: 1.3.5.1	4.8624	5.7332	4.7385	4.8726	0.1725	NS	NS	NS
EC:6.2.1.5	0.1001	0.1617	1.0779	1.5528	0.0212	**	*	NS
EC:5.4.99.2	4.8603	5.5625	4.8684	5.1543	0.0604	NS	*	NS
EC:6.4.1.3	3.4469	3.9384	3.4114	3.3103	0.0303	NS	NS	NS
EC:2.8.3.1	0.0161	0.0313	0.0022	0.0664	0.0002	*	**	**

^1^ BF(0), before feeding (0 h); ^2^ AF(4), after feeding (4 h); ^3^ SEM for feed × time; ^4^ NS, not significant (*p* > 0.05); *, (*p* < 0.05); **, (*p* < 0.01).

**Table 3 animals-10-00223-t003:** The percentage of reads mapped to each enzyme involved in propionate butyrate.

Enzyme	HF		HC		SEM ^3^	*p*-Value ^4^		
	BF0h ^1^	AF4h ^2^	BF0h	AF4h		Feed	Time	Feed × Time
EC: 1.2.7.1	0.0197	0.0144	0.5551	1.0831	0.0201	**	*	*
EC:2.3.1.9	0.2156	0.4360	0.1077	0.4766	0.0047	NS	*	NS
EC:1.1.1.157	0.4243	0.4258	0.3754	0.3273	0.0054	NS	NS	NS
EC:4.2.1.17	0.2757	0.2754	0.3051	0.2771	0.0033	NS	NS	NS
EC:1.3.8.1	0.8092	0.7743	0.2756	0.2281	0.0075	**	NS	NS
EC:2.8.3.8	0.4289	0.4697	0.3150	0.3494	0.0038	*	NS	NS
EC:4.2.1.55	0.0601	0.1067	0.0795	0.0992	0.0000	NS	**	*
EC:2.3.1.19	2.0545	2.2858	1.8426	2.0529	0.0127	*	*	NS
EC:2.7.2.7	1.6690	2.0296	2.1763	2.4183	0.0148	**	*	NS

^1^ BF(0), before feeding (0 h); ^2^ AF(4), after feeding (4 h); ^3^ SEM for feed × time; ^4^ NS, not significant (*P* > 0.05); *, (*P* < 0.05); **, (*P* < 0.01).

**Table 4 animals-10-00223-t004:** The relative expression (%) of main bacteria involved in the metabolism of fiber in rumen of cows fed two different diets.

Bacteria	HF		HC		SEM ^3^	*p*-Value ^4^		
	BF0h ^1^	AF4h ^2^	BF0h	AF4h		Feed	Time	Feed × Time
*Ruminococcus flavefaciens*	1.9712	2.6007	2.3312	2.6435	0.0421	*NS*	***	*NS*
*Ruminococcus albus*	0.1562	0.3060	0.3612	0.2482	0.0021	NS	NS	**
*Fibrobacter succinogenes*	0.3152	0.5240	0.1860	0.3638	0.0017	**	**	NS
*Butyrivibrio fibrisolvens*	0.0388	0.0180	0.0557	0.0672	0.000	**	NS	**
*Prevotella ruminicola*	3.8373	2.2827	2.9234	5.0617	0.0937	**	NS	**
*Selenomonas ruminantium*	0.8129	1.2319	0.6473	5.5426	0.0380	**	**	**
*Megasphaera elsdenii*	0.0099	0.0148	0.0022	0.0033	3.1E-6	**	NS	NS
*Veillonella alkalescens*	0.0140	0.0885	0.2692	0.1966	0.0010	**	NS	*

^1^ BF(0), before feeding (0 h); ^2^ AF(4), after feeding (4 h); ^3^ SEM for feed × time; ^4^ NS, not significant (*p* > 0.05); *, (*p* < 0.05); **, (*p* < 0.01).

## Data Availability

The Illumina sequencing raw data for our samples have been deposited in the NCBI Sequence Read Archive (SRA) under accession number: PRJNA522848.

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
