# Peer review of "Effects of High Forage/Concentrate Diet on Volatile Fatty Acid Production and the Microorganisms Involved in VFA Production in Cow Rumen"

_animals, 2020, doi:10.3390/ani10020223_

Round 1

Reviewer 1 Report

In my opinion this paper is excellent. In attached file only minor suggestions

Author Response

We are truly grateful to yours and reviewers’ critical comments and thoughtful suggestions for our manuscript. The suggestions of the reviewers were highly insightful and enabled us to improve the quality and significance of our manuscript. We thank the reviewers for their careful read and thoughtful comments on previous draft. Our point-by-point responses to each of the comments of the reviewers, please see the attachment

Reviewer 2 Report

L17 Change “shown” into “showed”

L22 Give more detailed interpretation of “time”

L48 Needless “[]”

L49 Add reference after “production”. Rewrite the sentence “During the before and after feeding”

L59 Add reference

L172-L175 The sentence is too long

L177 Add “which” before “is”

L.304 Change “can” into “could”

L 328 Add “that of” before “the”

Figure 1 and Figure 3 Please provide high definition figure

Author Response

We are truly grateful to yours and reviewers’ critical comments and thoughtful suggestions for our manuscript. The suggestions of the reviewers were highly insightful and enabled us to improve the quality and significance of our manuscript. We thank the reviewers for their careful read and thoughtful comments on previous draft. Our point-by-point responses to each of the comments of the reviewers, please see the attachment

Best regards

Yonggen zhang, Lijun Wang
